# The Effects of 3,5-Dimethylpyrazole on Soil Nitrification and Related Enzyme Activities in Brown Soil

Yuanchuang Lu [1,†] , Dongxiao Li [2,†], Changqing Li [1], Mengyu Sun [1], Zhijie Wu [3] and Zhimei Sun [1,*]

1 Key Laboratory for Farmland Eco-Environment of Hebei, College of Resources and Environmental Sciences, Hebei Agricultural University, Baoding 071000, China; yuanchuanglu2022@163.com (Y.L.); lixiaoqing19960815@163.com (C.L.); 13292258387@163.com (M.S.)
2 State Key Laboratory of North China Crop Improvement and Regulation, Key Laboratory of Crop Growth Regulation of Hebei Province, College of Agronomy, Hebei Agricultural University, Baoding 071000, China; nxyldx@hebau.edu.cn
3 Institute of Applied Ecology, Chinese Academy of Sciences, Shenyang 110016, China; wuzj@iae.ac.cn
* Correspondence: sunzhimei@hebau.edu.cn; Tel.: +86-13930892021
† These authors contributed equally to this work.

**Abstract:** Heterocyclic nitrogen compounds containing two adjacent nitrogen atoms generally have a significant effect on soil nitrification inhibition, and 3,5-dimethylpyrazole (DMP) is a typical representative of this structure. However, the inhibitory effect and the regulatory mechanism of DMP on soil N transformation are unclear. In this study, a microcosm with different concentrations of DMP was carried out in brown soil to detect the dynamic changes of soil $NH_4^+$–N, $NO_3$–N and related soil enzyme activities. Results showed that DMP inhibited soil nitrification effectively and decreased soil nitrate reductase activity, while increasing nitrite reductase and dehydrogenase activities. The inhibition effects were dose dependent, and DMP at the rate of 0.025 g kg$^{-1}$ dry soil showed the strongest inhibitory effect on N transformation in brown soil. The soil dehydrogenase activity was increased with an increasing DMP application rate. The changes in the soil's chemical and biological properties caused by DMP application provided a new idea for systematically explaining how DMP participated in the soil N transformation process. This study further implied that DMP would play positive roles in alleviating environmental pressure by delaying nitrate-N formation and decreasing the activity of nitrate reductase.

**Keywords:** 3,5-dimethylpyrazole; nitrification inhibition; nitrite reductase activity; nitrate reductase activity; dehydrogenase activity





## 1. Introduction

Most nitrogen (N) chemical fertilizers, either ammonium-N or ammonium-producing compounds such as urea, are usually oxidized to nitrate rapidly by nitrifying bacteria, especially in aerobic soil. However, nitrate is susceptible to leaching into groundwater, infiltrating surface runoff, or emitting into the atmosphere as nitrogen-based greenhouse gases through denitrification [1–3]. To minimize N loss and to limit N pollution, soil ammonia oxidation should be well controlled, without influencing crop production [4]. Nitrification inhibitors (NIs) are the compounds that can retard ammonium oxidation through inhibiting nitrifying microorganisms [5]. Many researches have shown that applying Nis could increase the size of the soil $NH_4^+$–N pool, enhance the bio-availability of soil $NH_4^+$–N, decrease the $NO_3^-$–N accumulation in soil, and ultimately alleviate the losses of fertilizer nitrogen [6–9]. In addition, NI could also decrease the application rate of nitrogenous fertilizer and improve N use efficiency, which may thus simplify N fertilization mainly through reducing the fertilization times and allowing greater flexibility in timing fertilization, thereby yielding both economic and environmental benefits [10]. Therefore, specific NIs are being increasingly recommended for intensive agriculture [11]. By now, a host of compounds,

both natural and synthetic, e.g., dicyandiamide, 2-amino-4-chloro-6-methylpyrimidine, 3,4-dimethylpyrazole phosphate, and 3-methylpyrazole-1-carboxamide, have been proven to significantly inhibit soil nitrification [12–15]. Among them, heterocyclic N-containing compounds are the main kinds of potential nitrification inhibitors with most containing two or three adjacent ring N atoms, e.g., 3,5-dimethyl-pyrazole-1-carboxamide, 4-bromo-3-methylpyrazole, and 3-aminopyrazole [13,14]. In recent years, nitrification inhibitors based on dimethylpyrazole have been widely used in agriculture, e.g., 3,4-dimethylpyrazole phosphate (DMPP) and 2-(3,4-dimethyl-1H-pyrazol-1-yl)-succinic acid (DMPSA), which have been shown to have the ability to inhibit nitrification [16,17]. DMPP has been reported to be efficient in regulating soil N transformation and influencing plant productivity, especially in alkaline soil [18]. The deep placement of urea combined with DMPP could reduce both emissions of $NH_3$ by 67% to 90% and $N_2O$ by 73% to 100%, respectively, to avoid N pollution swapping in cropland [19]. 3,5-dimethylpyrazole (DMP) was originally reported by others [13], who found it could inhibit soil nitrification when studying the relationship between the structure of heterocyclic nitrogen compounds and their nitrification inhibition effect. Since then, DMP has been mainly used as a sealer in automotive coatings and coil steel coatings to save energy, reduce production time, and reduce curing temperature (Jones, 2003); however, there have been no reports about this compound in the field of agriculture. Our previous work proved it had a better inhibitory effect than dicyandiamide [20]. A generally recommended rate (0.225–0.45 mg kg$^{-1}$) of DMP for moderate agro-climates is ineffective in inhibiting nitrification at high soil temperature [21]. In addition, high application rates of nitrification inhibitors should be avoided due to their phytotoxic and adverse effects on the activities of soil microflora. Up to now, little is known about its effects on urea-N transformation and related soil enzymatic activities. It is necessary to find the appropriate concentration of DMP, which could effectively inhibit nitrification in typical agro-ecological regions of China.

In this study, a microcosm was conducted to evaluate the effects of DMP application on urea-N transformation and related enzyme activities in brown soil, which would help us to understand the action mode of DMP on soil N transformation as well as its potential environmental effects.

## 2. Materials and Methods

### 2.1. Physical and Chemical Properties of DMP and Its Safety

DMP with a purity of 99% and a nitrogen content of 14.43% were provided by the Dalian Institute of chemistry, Chinese Academy of Sciences and Dalian University. The related structural formula of the compound is shown in Figure 1. The physical and chemical properties of DMP are shown in Table 1. The safety of DMP has been evaluated by the Liaoning Center for Disease Control, i.e., the LD50 of DMP is 1470 mg kg$^{-1}$, which means a low toxic substance through the acute oral toxicity test in mice.

**Table 1.** Physical and chemical properties of 3,5-dimethylpyrazole (DMP).

| (Item) | Properties |
|---|---|
| CAS Registry Number | 67-51-6 |
| Color and form | White or light yellow crystals |
| Molecular Formula | $C_5H_8N_2$ |
| Formula Weight | 96.13 |
| Melting point, Mp | 106–109 °C |
| Boiling point | 218 °C |
| Deblock temperature | 55–65 °C |
| Solubility | Soluble in water and acetone, easily soluble in ether and benzene |

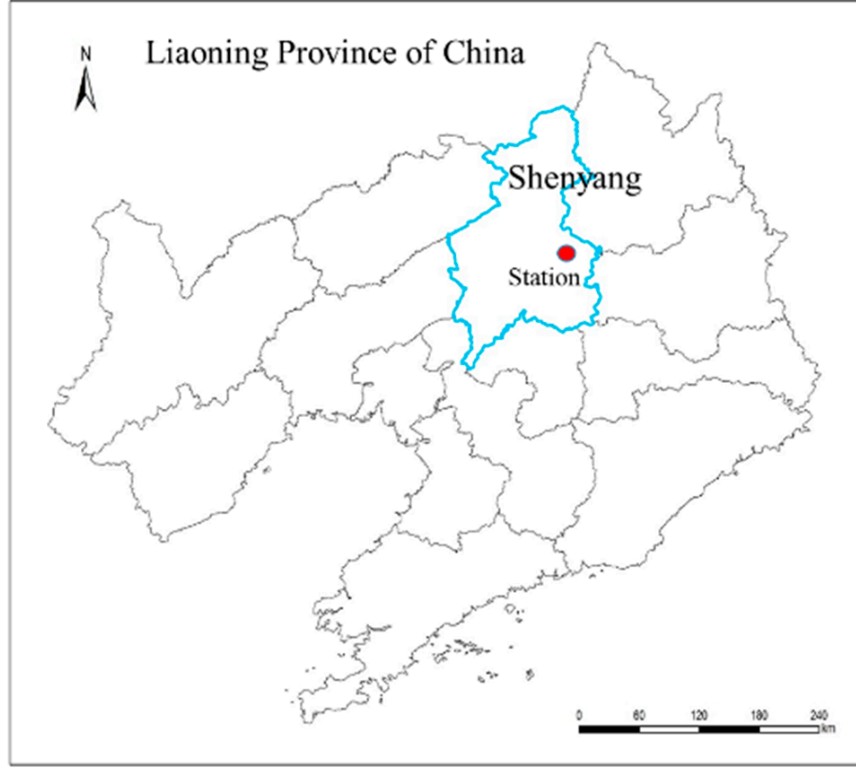

**Figure 1.** Chemical structure of DMP.

## 2.2. Study Site

Brown soil, a typical soil type in China, was collected for microcosm construction. Soil samples from the depth of 0–20 cm were collected from the Shenyang Experimental Station of Ecology (41°31′ N, 123°24′ E), Chinese Academy of Sciences, Liaoning Province, China (Figure 2). This site is a typical agricultural production area characterized by a warm, temperate, semi humid continental climate. The average annual temperature is 7–8 °C and the accumulated temperature ≥ 10 °C is 3300–3400 °C. The annual rainfall is 650–700 mm of which the frost-free period is 147–164 days with a single harvest per year. The basic physical and chemical properties of soil are shown in Table 2.

**Figure 2.** Location of Shenyang Experimental Station of Ecology, Chinese Academy of Sciences.

**Table 2.** Physical and chemical properties of tested soil.

| Soil Depth (cm) | Organic Matter (g kg$^{-1}$) | Total N (g kg$^{-1}$) | Available P (mg kg$^{-1}$) | Available K (mg kg$^{-1}$) | pH | Soil Texture (%) | | |
|---|---|---|---|---|---|---|---|---|
| | | | | | | Clay | Silt | Sand |
| 0–20 | 16.3 | 1.2 | 14.93 | 110.14 | 6.67 | 14.88 | 66.70 | 18.42 |

### 2.3. Microcosm Construction

The soil samples were air-dried, passed through a 2-mm sieve, moistened with distilled water to 50% water-holding capacity (WHC), and then pre-incubated at 25 °C–28 °C for 7 days. Then, four treatments with three replicates were designed including DMP0 (no DMP), DMP1 (0.0025 g kg$^{-1}$ dry soil), DMP2 (0.01 g kg$^{-1}$ dry soil), and DMP3 (0.025 g kg$^{-1}$ dry soil). In the meantime, 0.25 g urea N kg$^{-1}$ dry soil was added to each treatment. Throughout the incubation, the moisture content of the soil samples was adjusted to 60% of their water-holding capacity with distilled water and maintained at this level gravimetrically. The soil samples, equivalent to 500 g air-dried soil, were kept in half-sealed polythene bags so as to retain aerobic conditions and then incubated at 25 °C.

### 2.4. Measuring Methods

At 1, 3, 7, 14, 21, 35, 42, 56, and 70 days after the incubation, portions of soil from each treatment were collected to determine the following items. Soil pH was measured with an electric digital pH meter (pHS-3C, soil: water ratio, 1:2.5). Soil $NH_4^+$–N and $NO_3^-$–N were extracted by a 2 mol L$^{-1}$ KCl solution [22], and their contents were then determined by an AA3 Continuous Flow Analyzer (Bran—Luebbe Inc., Hamburg, Germany).

Soil urease activity and nitrate reductase activity were determined according to the method described by Tabatabai [23] and Kandeler [24], respectively. Soil nitrite reductase activity was expressed as the difference of $NO_2^-$–N concentration before and after incubation based on the modified methods of Muhammad et al. [25]. The method was detailed as follows: Soil samples were air-dried and passed through 1 mm sieve. Then, a 1 g soil sample was placed into a test tube (10 cm × 1.5 cm) and added with 20 mg $CaCO_3$, 2 mL 0.030 mol L$^{-1}$ $NaNO_2$, 2 mL 0.028 mol L$^{-1}$ glucose, and 1 mL distilled water. After that, the tubes were plugged with a rubber cap and mixed, then incubated at 30 °C for 24 h. At last, the extracted nitrite was determined calorimetrically at 520 nm with a developer.

Soil dehydrogenase activity was assayed by the 2,3,5-triphenyl tetrazolium chloride (TTC) reduction technique [23].

### 2.5. Calculation Methods of Nitrification Indexes

The following formulas are used to calculate the nitrification indexes.

$$\text{Apparent nitrification rate (\%)} = a/(a + b) \times 100\% \tag{1}$$

where a is soil nitrate nitrogen concentration and b is ammonium nitrogen content.

$$\text{Nitrification inhibition rate (\%)} = (A - B)/A \times 100\% \tag{2}$$

where A is the difference of nitrate nitrogen content before and after incubation with urea treatment alone and B is the difference of nitrate nitrogen content before and after incubation with urea and inhibitor treatment.

Calculation method of maximum nitrification rate ($k_{max}$) and retardation period ($t_d$)

The accumulation characteristics of $NO_3^-$–N with time (t) were quantitatively calculated according to Equation (3)

$$NO_3^-\text{–N} = \frac{a}{1 + (a/[NO_3^-\text{–N}]_0 - 1)\exp(-ak[t - t_0])} \tag{3}$$

where a and $[NO_3^--N]_0$ are the asymptotic and initial values of $NO_3^--N$, respectively, k is a constant, and $t_0$ is the initial time, which equals zero. The parameters a, k, and $[NO_3^--N]_0$ were calculated by the least-squares fit of Equation (3) to the experimental data of $NO_3^--N$ vs. t. In general, when the $NH_4^+-N$ content is not the factor of rate limiting, the maximal rate of nitrification will depend on soil properties, such as pH and soil temperature. Therefore, the maximal rate ($k_{max}$) was calculated as the maximal slope of equation

$$k_{max} = \frac{k \times a^2}{4} = \frac{ab}{4} \tag{4}$$

The delay period ($t_d$) was calculated as the value of t when the maximal slope was extrapolated to the initial value of $NO_3-N$

$$t_d = \frac{1}{ak} \ln\left[\frac{a}{(NO_3^--N)_0} - 1\right] + \frac{(NO_3^--N)_0 - a/2}{K_{max}} \tag{5}$$

To further evaluate the capacity of DMP for delaying soil nitrification, the daily ammonium loss rates were first calculated during the period in which soil ammonium decreased significantly by using the equation:

$$C_t = -k_t + C_0 \tag{6}$$

where $C_t$ is the soil ammonium concentration t days after the beginning of incubation, $C_0$ is the measured maximal ammonium concentration, and k is the calculated ammonium daily loss rate ($mg\,kg^{-1}\,day^{-1}$). Based on this, the ammonium half-life $T_{0.5}$, i.e., the time needed for the maximal soil ammonium concentration to be halved, was calculated using the following equation:

$$T_{0.5} = C_0/(2 \times k) \tag{7}$$

*2.6. Statistical Analysis*

The indices were the averages of three replicates, and they expressed on the basis of oven-dried soil (105 °C for 8–10 h), except that nitrite reductase activity was expressed on the basis of air-dried soil. Statistical analysis was performed by using one-way ANOVA processes of SPSS 22 software (v22.0, Chicago, IL, USA). Significant differences were determined as $p < 0.01$ using Duncan's multiple range test. Linear regression analyses were performed to find the significant relationships between soil properties and enzyme activities.

## 3. Results

### 3.1. Effects of Different Doses of DMP on Indicators of Soil Nitrification

The decrease of $NH_4^+-N$ and the accumulation of $NO_3^--N$ with time under different treatments are shown in Figure 3. For the treatment only applied with urea (DMP0), soil $NH_4^+-N$ content reached the maximum on day 3 of incubation and then decreased sharply, remaining at only 15.17% of the maximum value on day 7, which suggested that the oxidation rate of ammonium derived from urea was rapid in the absence of nitrification inhibitors. While treated with DMP at different concentrations, $NH_4^+-N$ contents in soil sample were significantly higher and $NO_3^--N$ contents were significantly lower than DMP0 ($p < 0.01$). On day 42 of incubation, the soil $NO_3^--N$ contents in DMP1 and DMP2 were nearly equivalent to that in DMP0, while that in DMP3 was lower by 60.16% and then up to almost the same value on day 70.

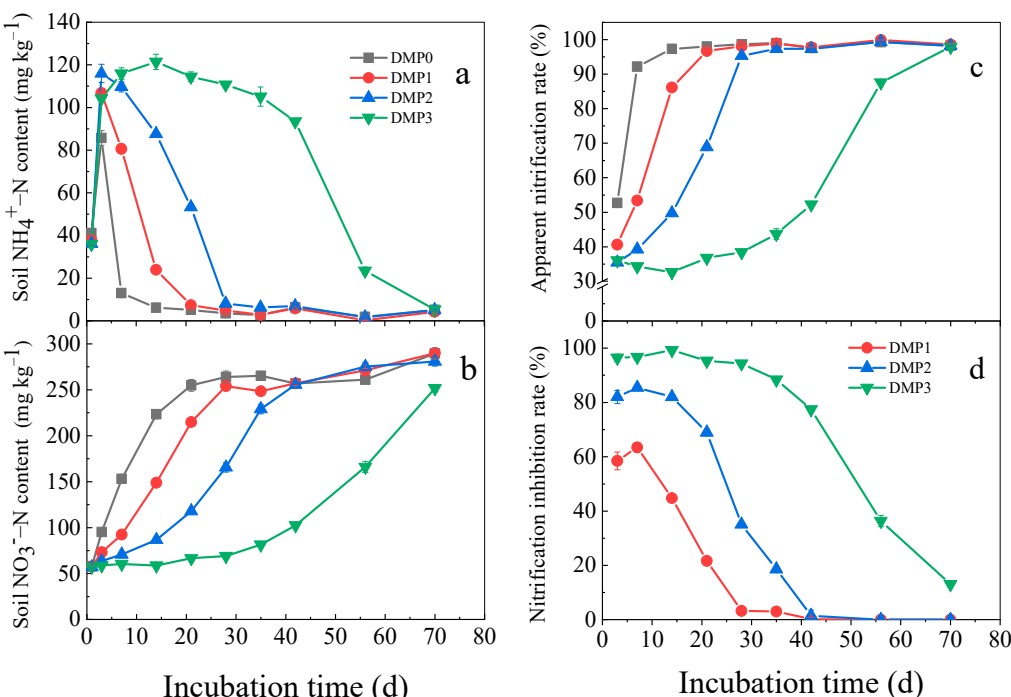

**Figure 3.** The dynamics of soil $NH_4^+$–N (**a**), $NO_3^-$–N (**b**) content, apparent nitrification rate (**c**), nitrification inhibition rate (**d**) under different doses of DMP. The error bars in figure indicate standard deviations (SD, *n* = 3). The same is as follows.

Apparent nitrification rate in all treatments showed an increasing trend. However, the time when reached to the maximum was significantly different. The time of reaching to the maximum in DMP0, DMP1, DMP2, and DMP3 was 7, 14, 35, and 70 days after incubation, respectively, which indicated that the apparent nitrification rate decreased significantly with the increasing of DMP. On the contrary, nitrification inhibition rate in DMP treatments showed a decreasing trend with a prolonged incubation period and an increasing application of DMP. The nitrification inhibition rate in DMP3 was the highest, significantly higher than that in DMP2 and DMP1. DMP2 was significantly higher than DMP1, and both decreased to 0 after 42 days. However, the nitrification inhibition rate in DMP3 still remained at approximately 15% on day 70 of incubation.

### 3.2. Fitted Equation of the Accumulation of Soil $NO_3^-$–N Content and the Variation of Soil Nitrification Parameters

Sigmoidal curves were found to be suitable for the accumulation trend of $NO_3^-$–N with time in DMP0, DMP1, and DMP2, which showed a delay phase, a maximal rate phase, and a retarded rate phase. While in DMP3 treatment, the accumulation trend of $NO_3^-$–N with time was described by the exponential curve (Table 3). The values of $K_{max}$ and $t_d$ in DMP0, DMP1, and DMP2 were calculated according to Equations (4) and (5). With an increasing DMP application dose, the $K_{max}$ value decreased significantly, while the $t_d$ value increased, suggesting that the inhibiting effect was significantly enhanced and the delay period was prolonged. It was noted that $K_{max}$ and $t_d$ in DMP3 were undetectable because DMP at a high concentration could alter the changing curve of the nitrification rate due to a profound inhibitory effect. The calculated result of the ammonium half-life $T_{0.5}$ is shown in Table 3. Compared with $T_{0.5}$ in DMP0, $T_{0.5}$ in DMP1, DMP2, and DMP3 were longer by 14.27%, 76.62%, and 205.64%, respectively.

**Table 3.** Fitted equation of the accumulation of soil $NO_3^- $–N content with time and the variation of soil nitrification parameters under different doses of DMP.

| Treatment | Fitted Equation | Correlation Coefficient | Maximal Rate $K_{max}$ (mg kg$^{-1}$ d$^{-1}$) | Delay Period $t_d$ (d) | Ammonium Half-Life $T_{0.5}$ (d) |
|---|---|---|---|---|---|
| DMP0 | $y = \dfrac{227.85}{1 + 4.61e^{-0.24t}}$ | 0.985 ** | 13.78 | 1.00 | 9.58 |
| DMP1 | $y = \dfrac{223.43}{1 + 5.68e^{0.15t}}$ | 0.998 ** | 8.48 | 2.21 | 13.10 |
| DMP2 | $y = \dfrac{232.26}{1 + 7.99e^{0.10t}}$ | 0.989 ** | 5.58 | 5.44 | 16.92 |
| DMP3 | $y = 48.30e^{0.02t}$ | 0.952 ** | - | - | 29.28 |

Note: $n$ = 11, $r_{0.05}$ = 0.602, $r_{0.01}$ = 0.735. $y$ is accumulation of $NO_3^-$–N and $t$ is incubation time. The level of significance in univariate ANOVA (analyses of variance) is denoted by ** ($p < 0.01$).

### 3.3. Effects of Different Doses of DMP on Soil Urease Activity

Soil urease is an obligate enzyme regulating the process of urea hydrolysis, which directly affects $NH_3$ volatilization and N use efficiency. Figure 4 showed soil urease activity decreased significantly with DMP dose increasing on the 1st day of incubation. Additionally, on the third day, urease activity reached the first peak value, with only DMP3 having a significantly lower value compared with the DMP0 treatment ($p < 0.05$). Then, it showed a "decrease-increase-decrease" changing trend with the second peak value appearing on day 35. Additionally, urease activity in DMP treatments was generally higher than those in DMP0 from days 7 to 35 of incubation. After day 35, this value decreased again with some significant differences observed between DMP treatments and DMP0 on days 56 and 70. This result indicated that DMP had a temporally inhibitory effect on soil urease activity during the hydrolysis process of urea.

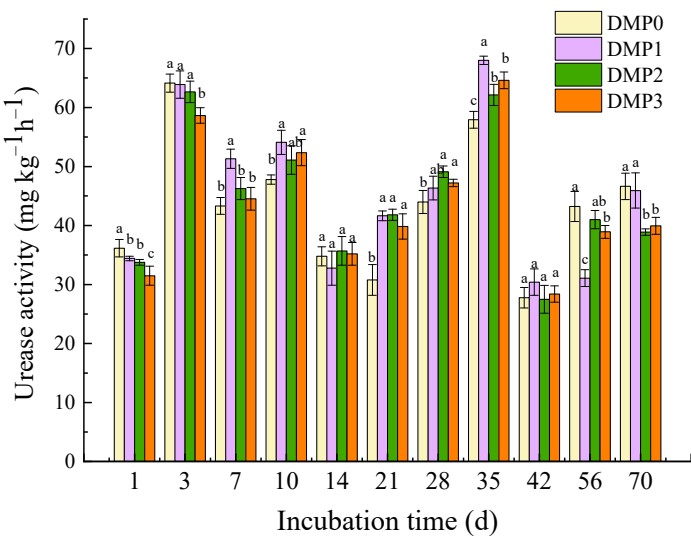

**Figure 4.** The dynamics of nitrate reductase activity under different doses of DMP. Different lower-case letters denote significant differences between different treatments for the same day ($p < 0.05$).

### 3.4. Effects of Different Doses of DMP on Soil Nitrate Reductase Activity

Nitrate reductase activity is a key enzyme regulating the transformation of nitrate-N to nitrite-N. Figure 5 shows that soil nitrate reductase activity was affected significantly by the addition of DMP. Compared with DMP0, soil nitrate reductase activity greatly increased on the third day, but it then decreased sharply down to the level of DMP0 on the tenth day. After 10 days of incubation, soil nitrate reductase activity decreased significantly,

inhibitory time was prolonged, and inhibitory effect was enhanced with the DMP dose increasing. This result indicated that DMP at an appropriate concentration could inhibit nitrate reductase activity effectively.

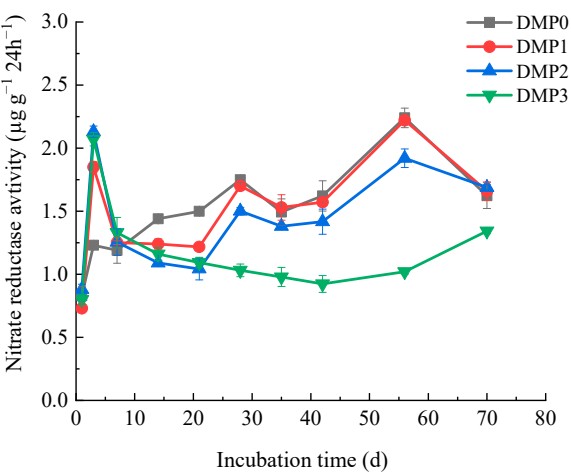

**Figure 5.** The dynamics of nitrate reductase activity under different doses of DMP.

*3.5. Effects of Different Doses of DMP on Soil Nitrite Reductase Activity*

Soil nitrite reductase is another key enzyme regulating nitrogen transformation. The dynamics of soil nitrite reductase activity with time was similar to that of nitrate reductase activity within the first 3 days (Figure 6), whereas a complete opposite trend was observed after 3 days. The nitrite reductase activity was decreased gradually from 3 days to 56 days, and it was significantly enhanced ($p < 0.01$) with the doses of DMP increasing. After 42 days incubation, there were no significant differences in nitrite reductase activity among DMP2, DMP1, and DMP0, but it was higher in DMP3 by 1.04 times than that in DMP0.

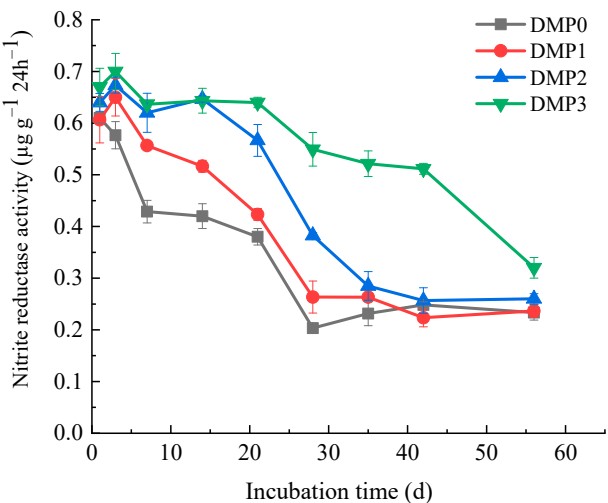

**Figure 6.** The dynamics of nitrite reductase activity under different doses of DMP.

*3.6. Effects of Different Doses of DMP on Soil Dehydrogenase Activity*

Soil dehydrogenase is a typical intracellular enzyme that directly indicates soil microbial growth and metabolism. Figure 7 shows that soil dehydrogenase activity (DHA) presented a peak on the third day. This value in DMP was significantly higher than that in DMP0 during days 3 to 21, and DHA increased significantly with the doses of DMP increasing ($p < 0.01$). After 3 days, DHA in all treatments showed a decreasing trend. Compared with DMP0 treatment, DHA in DMP3 was higher by 81.06% and 49.64% on 21 days and

42 days, respectively. In the later incubation period, DHA under DMP1 and DMP2 were also higher than DMP0, which was similar to the change of soil nitrite reductase activity. These results suggested DMP stimulated but did not affect total microbial activity.

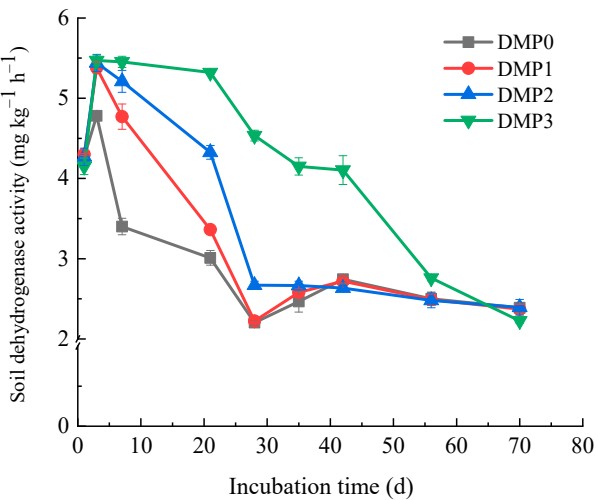

**Figure 7.** The dynamics of dehydrogenase activity under different doses of DMP.

### 3.7. Dynamic Changes of Soil pH

Soil pH, influencing directly soil enzyme activity and microorganism, was also significantly affected by the addition of DMP (Figure 8). Within the first 3 days, soil pH in all the treatments increased remarkably with urea hydrolysis, reaching a peak on the third day, then keeping a general decreasing trend. For the DMP3 treatment, soil pH was significantly higher than those in other treatments from 3–60 days of incubation, and it showed a slower decrease rate compared with that in DMP0. For DMP1 and DMP2, soil pH was also significantly higher than that in DMP0 during 3–30 days of incubation, then no significant differences were observed. In addition, the decreased rate of soil pH became slower with the increasing of DMP application rates. Soil pH at day 56 in DMP0 fell by 1.24 units compared with the initial pH of tested soil, while that in the DMP3 treatment was still greater by 0.91 units than that in DMP0. These results suggested that DMP application could increase soil pH effectively and maintain a high level for a long time.

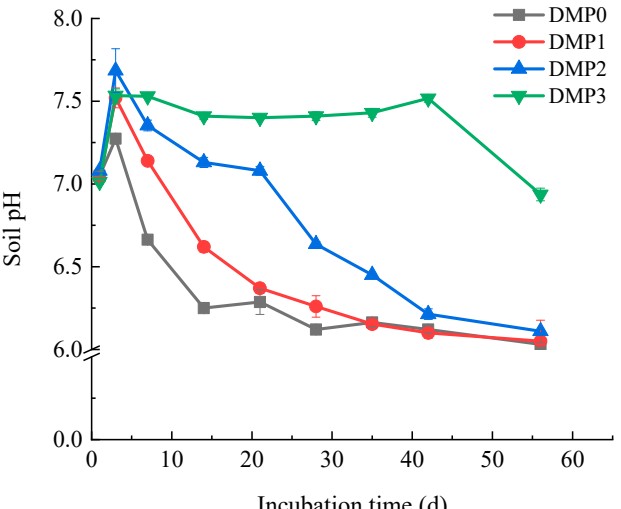

**Figure 8.** The dynamics of soil pH under different doses of DMP.

### 3.8. Relationships between Soil Chemical Properties and Biological Activities

There was a significant linear relationship between soil chemical properties and enzyme activities (Table 4). Here, soil $NH_4^+$–N content had significant positive correlations with soil enzyme activities and soil pH, while soil $NO_3^-$–N content had negative correlations with soil enzyme activities except soil nitrate reductase activities. Soil pH was positively correlated with soil nitrite reductase activities and DHA, but negatively correlated with soil nitrate reductase activity. These results suggested that the detected three soil enzyme activities may be strongly affected by the changes of soil $NH_4^+$–N, $NO_3^-$–N content, and soil pH.

**Table 4.** Linear relationship coefficients between soil physical–chemical properties and soil biological parameters (df = 26).

| Soil Properties | $NH_4^+$–N (mg kg$^{-1}$) | $NO_3^-$–N (mg kg$^{-1}$) | pH |
|---|---|---|---|
| nitrate reductase activity(ug N g$^{-1}$·24 h$^{-1}$) | 0.5094 ** | 0.8377 ** | −0.6377 ** |
| nitrite reductase activity (μg N g$^{-1}$·24 h$^{-1}$) | 0.7715 ** | −0.9576 ** | 0.8464 ** |
| DHA (mg N kg$^{-1}$·h$^{-1}$) | 0.6968 ** | −0.7556 ** | 0.7726 ** |
| pH | 0.8859 ** | −0.8840 ** | |

** significant at $p < 0.01$ levels.

## 4. Discussion

Soil $NH_4^+$–N is favored to be adsorbed by soil colloids and assimilated by plants and microbes, while soil $NO_3^-$–N is easily leached out or denitrified accordingly, causing potential adverse effects on environmental quality and public health [26]. Therefore, how to retard the nitrification of soil $NH_4^+$–N has been paid more attention in recent years. Our study showed that the application of DMP could significantly increase the $NH_4^+$–N content while decreasing the $NO_3^-$–N accumulation in soil, for which the effects were increased with an increasing DMP dose. The inhibitory efficiency of different doses of DMP on soil nitrification could be observed more intuitively from our determination of soil nitrification potential, indicating an active nitrifier population size [27]. The lowest soil nitrification potential found in DMP3 illustrated that an application of 0.025 g DMP kg$^{-1}$ dry soil could inhibit soil nitrifier activity more efficiently than other treatments within a long period. The gradual increase of soil nitrification potential under a lower dose of DMP was possibly due to their degradation, which not only de-repressed soil nitrifiers but also provided available energy sources for the nitrifier population and other microorganisms [28].

Soil nitrate- and nitrite reductases, which catalyze the reduction of $NO_3^-$ to $NO_2^-$ and $NO_2^-$ to $N_2O$ or $NH_4^+$, respectively, are the two important enzymes involved in the three processes of soil denitrification and dissimilatory and assimilatory reduction of $NO_3^-$ to $NH_4^+$ [29,30]. The assimilatory reduction of $NO_3^-$ to $NH_4^+$ was often neglected because this process was strongly inhibited by glutamine formed though microbial assimilation of $NH_4^+$; while the other two processes had been paid broad attention and mainly studied by measuring $N_2$ and $N_2O$ production with labeled $NH_4^+$–N derived from added $^{15}NO_3^-$–N [31,32]. Our study showed that DMP application had a negative effect on soil nitrate reductase activity and a positive effect on soil nitrite reductase activity, and the effects increased with increasing DMP doses. This result might not only be related to the changes of soil microbial activity but also the variation of soil mineral N and pH caused by DMP application. The lower soil nitrate reductase activity after applying DMP attributed to the low process of denitrification and the dissimilatory reduction of $NO_3^-$–N to $NO_2^-$–N might be advantageous to the dissimilatory reduction of $NO_2^-$–N to $NH_4^+$–N with a higher nitrite reductase activity. A higher C/$NO_3^-$–N ratio was reported to be more favorable for the process of dissimilatory $NO_3^-$–N reduction to $NH_4^+$–N (DNRA) than the denitrification process [33–35]. Yin et al. [36] reported that soil labile carbon was the key factor influencing the partitioning of nitrate reduction between denitrification and DNRA, and glucose addition could significantly increase $NH_4^+$–N content derived from DNRA by

20%~50% compared with non-glucose treatments. Therefore, the higher labile organic C estimated by the $CO_2$ production under DMP was probably additional evidence that DMP will be more favorable for the process of DNRA than denitrification [37].

N$_2$O and N$_2$ were produced not only by denitrifying bacteria but also by ammonia-oxidizing bacteria, such as Nitrosomonas europaea [38,39]. N. europaea nitrite reductase is similar to denitrifying nitrite reductase, and it shares a nitrite-reducing mechanism with classical denitrification [40,41]. The inhibition of DMP on soil ammonia−oxidizers must decrease the flux of greenhouse gases through nitrification. Furthermore, Cooper and Smith [42] showed that the limitation of reduction of $NO_3^-$ caused by the lower Nitrate reductase activity and the lower $NO_3^-$–N accumulation in DMP is the main rate−limiting step for denitrification in acid soils. In this study, urease activity also decreased significantly with DMP during the initial 3 days, a peak period of urea hydrolysis, which have brought to delay this hydrolysis, avoiding the formation of high concentration ammonium, and contributing to less ammonia volatilization loss. These might be the primary reason why applying nitrification inhibitors, e.g., DCD, DMPP, and nitrapyrin, could decrease the flux of greenhouse gases [43–45].

Because soil dehydrogenase is exclusively an intracellular enzyme and links to viable cells [46], it is usually considered as an indicator of the oxidative metabolism and microbiological activity in soils, which may also indicate the availability of soil C for soil microorganisms [47]. In this study, the higher soil dehydrogenase activity in the DMP treatment than in the control demonstrated that DMP could stimulate soil microbial activity, which might be due to more available C released with the degradation of DMP and/or a changing pH. However, we didn't find a significant relationship between DHA and soil total organic C (data not shown), and several other studies have also shown that DHA was poorly correlated with soil organic matter [48–50]. It was speculated that more available C could be consumed as an energy source by other kinds of soil microorganisms, resulting in increased microbial activity. As known, the most favorable ratio of C/N is 20−30:1 for microorganisms, but there was only approximately 7.88:1 in our soil samples (Table 2). So, the soil was restricted by carbon, and total microbial activity was inactive; but, it would be activated and increased once trace carbon was added from DMP [51]. Additionally, other significant impact factors on soil pH that influenced microbial community and microbial activities directly should be paid more attention in the future [52]. Therefore, we propose a potential model for the inhibiting effect and the partial mechanism of DMP on the nitrification process that further results in beneficial environmental changes and crop production (Figure 9).

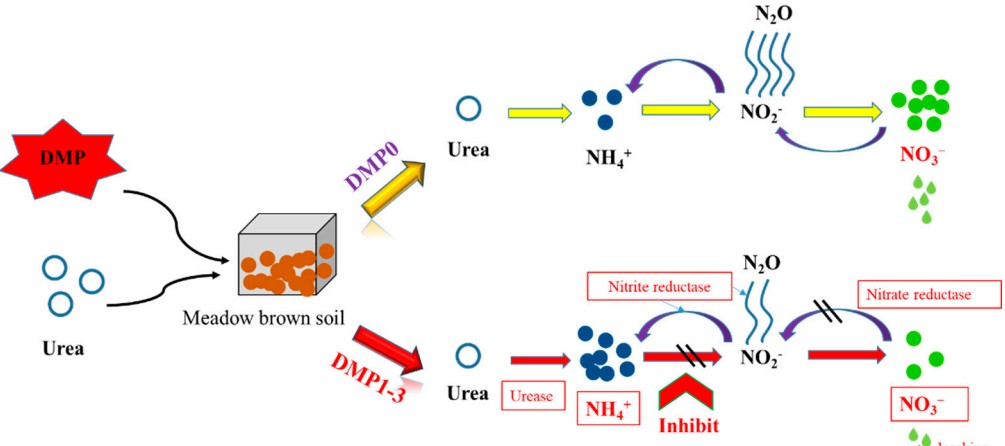

**Figure 9.** A work model of DMP on soil nitrification. Three inhibitory treatments are represented by red arrows, only urea is represented by yellow arrows, and the chemical reaction process is inhibited by double diagonals.

## 5. Conclusions

DMP could inhibit soil nitrification effectively. It decreased the soil nitrate content and the soil nitrate reductase activity, which means urea combined with DMP application could reduce $NO_3^-$ leaching and nitrogen gaseous emission. DMP increased the soil nitrite reductase activity and therefore increased the dissimilatory reduction of $NO_3^-$ to $NH_4^+$. Moreover, DMP also stimulated soil microbial activity, especially when the DMP dose reached 0.025 g $kg^{-1}$ dry soil. The results provided a new idea for systematically explaining how DMP participated in soil nitrogen transformation. DMP would contribute to solving the low utilization rate of nitrogen fertilizer and realizing green and low-carbon agricultural production.

**Author Contributions:** Methodology, Z.S.; software, Y.L.; formal analysis, D.L. and Z.S.; investigation, C.L. and M.S.; data curation, Z.W.; writing—original draft preparation, Z.S. and Y.L.; writing—review and editing, D.L. and Z.S.; supervision, Z.S. All authors have read and agreed to the published version of the manuscript.

**Funding:** This work was financially supported by the National Key Research and Development Program of China (2021YFD1901004), the Science and Technology Project of Hebei Education Department (QN2020147).

**Institutional Review Board Statement:** Not applicable.

**Informed Consent Statement:** Not applicable.

**Data Availability Statement:** Not applicable.

**Conflicts of Interest:** The authors declare no conflict of interest.

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
