# Peer review of "The Effects of 3,5-Dimethylpyrazole on Soil Nitrification and Related Enzyme Activities in Brown Soil"

_agronomy, doi:10.3390/agronomy12061425_

Round 1

Reviewer 1 Report

Manuscript Number: agronomy-1724745

Title: “Effects of 3,5-dimethyl pyrazole on soil nitrification and related 2 enzyme activities in Northeast China” by Yuanchuang Lu et al.

The authors have developed nitrogen heterocyclic 3, 5-dimethyl pyrazole (DMP), an organic compound for efficient and stable fertilizing activity. The necessary experimentation was done and the supportive results are analyzed.

Though the authors have tried to arrive at the significant results based on their defined objectives, still there are many points to be addressed. The following are the comments which must be considered and to be addressed by the authors to improve the quality of the manuscript.

  1. Reframe the abstract with important highlights of the work.
  2. Improve the resolution of the figures to reach good quality.
  3. Try to redraw figure 2.
  4. The synthesis of the DMP should be clearly discussed with proper literature references.
  5. And the structural characterizations are needed to be performed.
  6. For the dynamics of the soil content, only plots are given in the manuscript. The data table should be provided along with plots.
  7. In table 2, what exactly the significance of the correlation coefficient, discuss.
  8. Sections 3.3 to 3.7 should be clearer with the significance of the results.
  9. Have you tried crystallization of the compounds?
  10. Provide at least powder X-ray data for the compounds with cell parameters and crystallite size.
  11. Any other characterization for interaction studies must be provided.
  12. The conclusion must be reframed with the significant findings of the studies.
  13. What are the significant results of the studies of the dynamics?
  14. The overall manuscript should be improved in all the sections. And should follow the journal guidelines.

With all the above clear comments, I strongly recommend the revision of the manuscript before it can be published. After significant improvement, the manuscript can be considered for publication.

Reviewer 2 Report

Manuscript ID: agronomy-1724745

Title: Effects of 3,5-dimethylpyrazole on soil nitrification and related enzyme activities in Northeast China

 Thank you very much for the interesting and contemporary and practical article. The authors of the manuscript took a good topic.  Generally, the manuscript is written in the scope of the journal. The manuscript needs some corrections.

(1) Line 15: Add "the" before "transformation "

(2) Line 22: Change "improving" to "improve"

(3) Line 26: Add "the" before " soil "

(4) Improve the introduction section, providing the problem

(5) Line 36: Add "the" before " atmosphere "

(6) Lines 37, 153: Delete " In order "

(7) Line 76: Change "is" to "were"

(8) Line 79: Change " have " to " has "

(9) Lines 139, 144: Change " was " to "were"

(10) Line 164: Change " All the " to " All" or "The "

(11) Line 199: Add "the" before " exponential curve "

(12) Line 272: Add "a" before " high level "

(13) Improve the 4. Discussion section

(14) Line 298: Delete " which "

(15) Line 350: Add "an" before " energy source "

(16) Line 354: Add "was" before " added "

(17) Line 356: Delete " by "

(18) Line 370: Change " indicated " to " indicate"

(19) In figure 6: Change " Soil Sehydrogenase " to " Soil dehydrogenase "

(20) It is preferable to shorten the "5. Conclusions"

References

(21) Journal name is italic, Please revise lines 390 (

Soil Biol. Biochem.), 393 (Plant Soil. ), ..etc.…

(22) Only the first letter in Journal name is capital. Such as Line 429: Change "J ENVIRON MANAGE " to "J Environ Manage "; Line 433: Change " EUR. J. AGRON." to " Eur. J. Agron."; Line 517: Change " CARBON MANAG." to " Carbon Manag."

Author Response

Dear reviewer:

Thank you for your useful comments. We have revised our manuscript accordingly.  Responses to reviewer’ comments are provided below.

  1. Line 15: Add "the" before "transformation "

Response: Thanks for reviewer’s comment. We have reframed the abstract. Please see line 17 (Line numbers from the clean manuscript. The same below).

  1. Line 22: Change "improving" to "improve"

Response: Thanks for your comment. We have reframed the abstract. Please see line 14-27.

  1. Line 26: Add "the" before " soil "

Response: We have added "the". Please see line 25.

  1. Improve the introduction section, providing the problem

Response: Thanks for your comment. We have improved the introduction section and provided the problem. Please see lines 69-72.

  1. Line 36: Add "the" before " atmosphere "

Response: We have added "the". Please see line 35.

  1. Lines 37, 153: Delete " In order "

Response: We have deleted " In order ".

  1. Line 76: Change "is" to "were"

Response: We have changed "is" to "were". Please see line 79.

  1. Line 79: Change " have " to " has "

Response: We have changed "have" to "has". Please see line 82.

  1. Lines 139, 144: Change " was " to "were"

Response: We have changed "was" to "were". Please see lines 137 and 142.

  1. Line 164: Change " All the " to " All" or "The "

Response: We have changed "All the" to "The". Please see line 161.

  1. Line 199: Add "the" before " exponential curve "

Response: We have added "the" before " exponential curve ". Please see line 198.

  1. Line 272: Add "a" before " high level "

Response: We have We have reframed this section. Please see line 263.

  1. Improve the 4. Discussion section

Response: We have improved the “4. Discussion section”. Please see line 290

  1. Line 298: Delete " which "

Response: We have deleted " which ".

  1. Line 350: Add "an" before " energy source "

Response: We have added "an" before " energy source ". Please see line 348.

  1. Line 354: Add "was" before " added "

Response: We have added "was" before " added ". Please see line 352.

  1. Line 356: Delete " by "

Response: We have deleted " by ".

  1. Line 370: Change " indicated " to " indicate"

Response: We have changed " indicated " to " provide ". Please see line 368.

  1. In figure 6: Change " Soil Sehydrogenase " to " Soil dehydrogenase "

Response: Thanks for your comment. We have changed " Soil Sehydrogenase " to " Soil dehydrogenase ". The modified picture is shown in Figure 6

  1. It is preferable to shorten the "5. Conclusions"

Response: Thanks for your comment. We have reframed the conclusion with the significant findings of the studies. Please see lines 363-371.

  1. Journal name is italic, Please revise lines 390 (Soil Biol. Biochem.), 393 (Plant Soil.), .etc.…

Response: Thank you for pointing out the format problem. We have revised all the journal names to italic. Please see lines 386, 389 and 431.

  1. Only the first letter in Journal name is capital. Such as Line 429: Change "J ENVIRON MANAGE " to "J Environ Manage "; Line 433: Change " EUR. J. AGRON." to " Eur. J. Agron."; Line 517: Change " CARBON MANAG." to " Carbon Manag."

Response: Thank you for pointing out the format problem. We have revised all the journal names. Please see lines 425, 428, 442, 512.

Special thanks to you for your good comments.